# AutoSpineAI: Lightweight Multimodal CAD Framework for Lumbar Spine MRI Assessments

Saied Salem [†]
*Department of Artificial Intelligence, College of AI Convergence, Daeyang AI Center, Sejong University,*
Seoul 05006, Korea
ORCID: 0009-0002-8948-2167

Afnan Habib
*Department of Artificial Intelligence and Data Science, College of AI Convergence, Sejong University,*
Seoul 05006, Korea
ORCID: 0009-0001-7609-5523

Mukhlis Raza
*Department of Artificial Intelligence, College of AI Convergence, Daeyang AI Center, Sejong University,*
Seoul 05006, Korea
ORCID: 0009-0001-7704-5149

Zaid Al-Huda
*Stirling College Chengdu University*
Chengdu 610106, P. R. China
ORCID: 0000-0002-8920-7635

Omar Al-maqtari
*School of Computing and Artificial Intelligence Southwest Jiaotong University*
Chengdu 610106, P. R. China
ORCID: 0000-0002-3200-4781

Bilal Ertuğrul
*Department of Neurosurgery, Faculty of Medicine, Fırat University,*
Elazığ 23119, Türkiye
ORCID: 0000-0001-7812-3332

Özal Yıldırım
*Department of Software Engineering, Technology Faculty, Firat University,*
Elazığ 23119, Türkiye
ORCID: 0000-0001-5375-3012

Yeong Hyeon Gu
*Department of Artificial Intelligence and Data Science, College of AI Convergence, Sejong University*
Seoul 05006, Korea
ORCID: 0000-0002-4457-4407

Mugahed A. Al-antari[*,†,], *IEEE Member*
*Department of Artificial Intelligence and Data Science, College of AI Convergence, Sejong University*
Seoul 05006, Korea
ORCID: 0000-0002-4457-4407

*Abstract*—**Automated spine lumbar MRI analysis improves clinical workflow and diagnostic accuracy for lumbar spinal stenosis (LSS). In this paper, we introduce *AutoSpineAI*, a novel fully automated CAD framework for lumbar spine MRI analysis and structured medical report generation (sMRG) leveraging large language models (LLMs). The system processes 3D MRI DICOM volumes by extracting mid-sagittal slices for vertebrae and intervertebral discs (IVDs) segmentation and localizes corresponding axial slices using 3D cross-projection algorithm. For sagittal and axial slices segmentation, a novel lightweight efficient compact model (ECM) is proposed by integrating multi-attention mechanisms within a compact AI architecture to extract the quantitative spinal structural measurements (SSM): disc degeneration, vertebral anomalies, and other alignment irregularities. These structured measurements and assessments are integrated and merged in prompts for a novel hybrid agentic LLM-driven retrieval system that combines semantic information and knowledge graph-based reasoning to generate detailed level-wise diagnostic report: vertebrae and IVDs. *AutoSpineAI* achieves Dice scores of 97.58% and 94.01% for sagittal and axial segmentation, respectively, and generates a structured full report by Gemma3 LLM within 30 seconds per patient, achieving 83.51% Bert F1-score, 19.33% Meteor, and 15.31% Rouge1. *AutoSpineAI* seems to be a scalable and interpretable for clinical and practical solutions for MRI LSS.**

*Keywords*—**Lumbar spinal stenosis (LSS); Computer-aided Diagnosis (CAD); Hybrid Agentic RAG; Large Language Model (LLM); Structured Medical Report Generation (sMRG).**

## I. INTRODUCTION

Chronic low back pain (CLBP) is recognized as a major public health issue, consistently ranking among the leading causes of disability worldwide [1]. One of the principal contributors to CLBP is lumbar degenerative spine diseases, which affect approximately 266 million individuals annually [2]. Spinal alignment disorders like vertebral deformities have shown significant prevalence, with an estimated 44 million cases reported in the USA alone in 2017 [3]. These conditions often lead to progressive mobility impairments, underscoring the need for accurate and efficient diagnostic techniques [4]. Magnetic resonance imaging (MRI) serves as the gold standard for the non-invasive spinal pathologies assessment offering detailed visualization of intervertebral discs, vertebrae, and spinal alignment under standardized imaging protocols [5]. To mitigate this, AI-powered CAD systems have been introduced, demonstrating notable success in enhancing diagnostic performance and reproducibility [6]. Generating accurate and comprehensive assistive radiology reports continues to be significant challenges in clinical workflows. In this study, we introduce *AutoSpineAI*, an innovative CAD system that automates the process of diagnosing lumbar spine diseases and generating full structured vertebra and IVD level-wise medical reports. The framework processes multimodal data including 3D sagittal and axial MRI DICOMs and the corresponding clinical text reports, leveraging state-of-the-art LLMs to produce accurate and contextualized diagnostic reports. The major innovations of this work are summarized as follows,

(1) A novel end-to-end CAD system is proposed to analyze MRI sagittal and axial imaging data with the associated clinical text reports towards automatic multimodal spinal assessment and full medical report Generation (sMRG).

(2) Lightweight segmentation model is designed to extract critical spine anatomical structures for LSS diagnosis.

(3) A 3D axial cross-projection mechanism is introduced to automatically localize axial slices corresponding to the mid of each IVDs in sagittal view.

(4) Disease-specific spinal measurements are derived from sagittal and axial segmentation outcomes to support LSS assessment and for accurate structured report generation.

(5) A novel hybrid agentic LLM-based RAG module is proposed, combining semantic information and domain knowledge structured disease relations to enhance retrieval for clinically accurate reports generation.

## II. RELATED WORK

Recent AI advancements have enhanced CAD systems for medical imaging, enabling automated diagnostics and report generation to streamlines clinical workflow [7]. Al-kafri et al.

---

* Corresponding Author. † Equal contribution to this work.

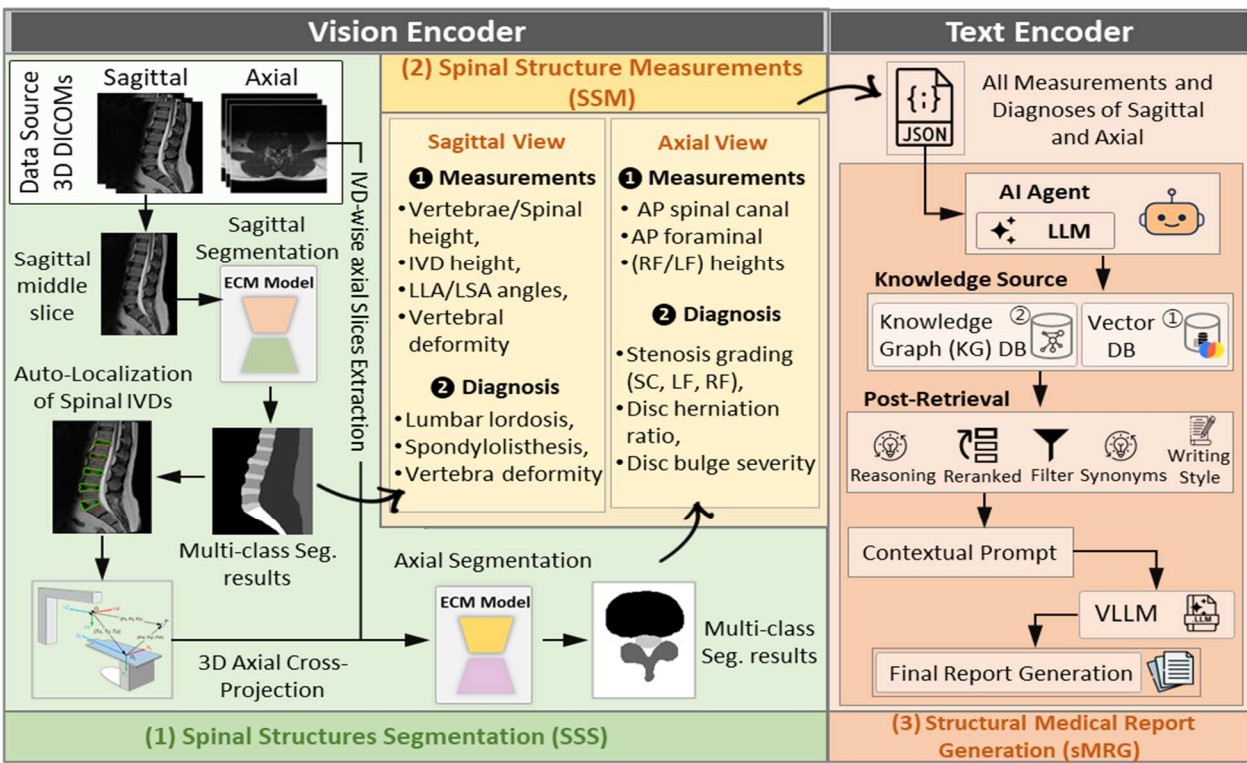

Fig. 1. End-to-end structure of the proposed CAD system including main three components: (1) Spinal Structures Segmentation (SSS), (2) Spinal Structure Measurements (SSM), and (3) Structural Medical Report Generation (sMRG).

[8] focused on segmentation key axial spine structures using SegNet, but lacked downstream diagnostic tasks. Later Studies by Natalia et al. [9], Masood et al. [10], Zheng et al. (2022) [11], and Al-Haidri et al. (2025) [12] integrated quantitative spinal measurements following segmentation to enable disorder detection and classification. While these approaches improved diagnostic specificity, they often faced challenges in generating detailed and clinically useful medical reports. Yi et al. (2024) introduced TSGET, a two-stage global enhancement layer that leverages the image-level global features to generate detailed clinical reports. Yu et al. (2025) [13] proposed a lightweight AI framework for classification and text report generation using the MIMIC-CXR chest X-ray and bladder pathology datasets. To our knowledge, this is the first work to focus on detailed report generation for MRI-based lumbar spine diseases. The proposed CAD system combines a vision encoder (i.e., lightweight segmentation and extraction of spinal measurements) with a text encoder to generate comprehensive structured medical reports (sMRG), advancing automation in lumbar spine diagnosis and evaluation.

## III. METHODOLOGY

Fig. 1 depicts the proposed AutoSpineAI which integrates vision and text encoders to generate high-quality multimodal image and text embeddings. The proposed lightweight segmentation model extracts key anatomical spine structures from sagittal and axial views. It begins by segmenting the mid-sagittal slice to identify lumbar IVDs, because it consistently provides the clearest anatomical representation of the vertebral bodies, intervertebral discs, and spinal canal. This central slice is a reliable reference across patients; it captures key structures required for accurate segmentation. Then, 3D cross-projection to align the corresponding axial images.

Using the segmented regions from both sagittal and axial slices, pathological spinal indices are computed and stored in an encrypted JSON file and passed into the structured prompt for LLM-based report generation. The proposed AgenticRAG, retrieves the *top-k* similar documents from vectorDB, and disease relationships from the temporal graph DB. The proposed hybrid semantic knowledge and structured disease relations could enhance the relevance of medical report generation. The primary objective here is to combine the vision encoder's outcomes to be explained within the sMRG, ensuring the balance of clarity among clinical accuracy, diagnostic clarity, and computational efficiency.

### A. Dataset

We use a publicly available lumbar spine MRI dataset [14]. It consists of paired T2 (sagittal) and T1(axial) MRI 3D DICOM volumes from 515 patients with symptomatic low back pain, and corresponding clinical findings reports. The dataset contains a total of 48,345 MRI slices, with image resolutions mostly at 320×320 pixels and 12-bit pixel precision. Expert neurologists manually annotated the mid-slice sagittal and some selected axial images. For the sagittal view [15], the annotations were derived based on the mid-slice to segment the vertebrae with their IVDs in between, sacrum, posterior elements (PosteriorA/PosteriorB), and anterior spinal structures. For the axial view [16], the annotations delineate four regions: IVD, posterior element (PE), thecal sac (TS), and the area between the anterior and posterior (AAP). For the ablation study, we use 100 cases (real private dataset).

### B. Proposed AI-based CAD Framework

#### 1) Contour extraction module

Fig. 2 depicts the proposed segmentation lightweight efficient compact model (ECM). Fig.2(a) shows the backbone comprises an input convolutional layer followed by four cascaded encoder blocks (E-blocks) in a (2,3,4,3)

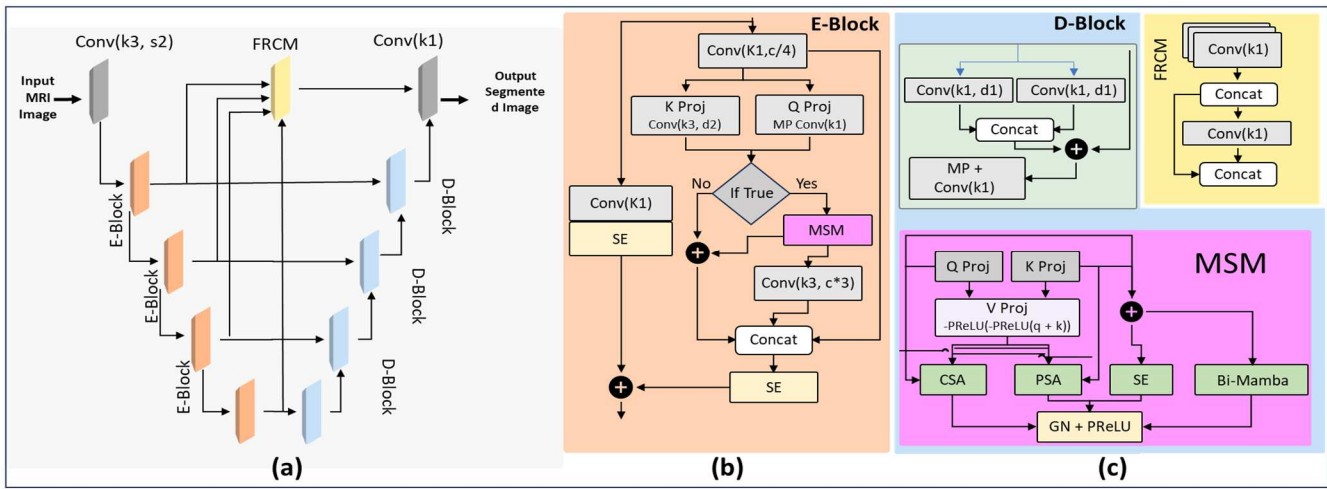

Fig. 2. The proposed segmentation lightweight model (i.e., efficient compact model (ECM)). (a) Overall encoder-decoder network, (b) E-Block structure with channel squeezing, K/Q projections, and Squeeze-and-Excitation (SE) blocks, abd (c) D-Block structure with parallel convolution paths and feature fusion, alongside Multi-Scale Module (MSM) with multiple attention mechanisms.

configuration. Each E-block, detailed in Fig. 2(b), employs a dimensional reduction strategy, where input features are compressed to one-fourth of their original dimension through a convolution layer. These compressed representations subsequently undergo parallel transformations via dilated convolutions and max pooling to generate Query (Q) and Key (K) projections. The Q-K fusion is enhanced through an additional convolutional layer, with residual connections preserving gradient flow. Attention mechanisms are hierarchically integrated from the second encoder stage onward. A non-linear approximation technique for Value (V) projection is implemented through dual parametric ReLU activations operating on positive and negative Q-K fusion values. The resulting Q, K, and V projections feed into our multi-attention framework comprising: (1) Channel Self-Attention (CSA) that captures inter-channel dependencies; (2) Patch Self-Attention (PSA) that models spatial context across feature patches; (3) Squeeze-and-Excitation (SE) that recalibrates channel-wise feature responses; and (4) Bidirectional Mamba (Bi-Mamba) that efficiently captures long-range dependencies along the channel dimension. The decoder pathway consists of four stages, each containing a decoder block (D-block), followed by a Feature Refinement and context module (FRCM) and an output convolution layer. Both D-blocks and FRCM utilize dilated convolutions at varying rates to expand the effective receptive field. Max-pooling operations after each encoder stage and corresponding up-sampling in the decoder facilitate multi-scale feature learning. Dropout regularization (p=0.2) is incorporated to mitigate overfitting.

*2) 3D Axial projection allocates the axial slices:*
After segmenting the sagittal IVDs, the centroid of each IVD contour is projected into 3D acquisition space using DICOM metadata. This enables accurate localization of corresponding axial slices that intersect the midline of each IVDs, ensuring targeted axial MRI extraction.

*3) Spinal Structure Measurements (SSM)*
A set of predefined clinically relevant measurements is quantified from extracted pathological structures' contours in each sagittal and axial view, to support spinal disease assessment and preoperative planning. The measurements were derived by image processing algorithms (tools) from both views. From a sagittal view, inspired by Masood et al. (2022)[11], lumbar structural measurements, including vertebrae height, each IVD or disc height, two angles (i.e., lumbosacral angles (LLA) and lumbar lordotic angle (LSA), and spinal height. Additionally, three different spinal disorders are identified: lumbar lordosis (normal, hyper lordosis, and hypo lordosis), spondylolisthesis (normal, posterior displacement, or anterior displacement), and the degree of vertebra deformation. Vertebral deformity is assessed using the Genant grading system [17], providing four vertebral gradings for wedge and biconcave deformities (normal, mild, moderate, and severe). From the axial view, the anterior-posterior (AP) distance in mm is measured among the segmented contours of IVD and PE. The axial measurements serve for identifying stenosis grading (i.e., normal, mild, moderate, or severe) for spinal canals, left foraminal (LF), and right foraminal (RF) regions. The CAD system determines the herniation level (i.e., minor or severe) based on the calculated herniation ratio presented in [10].

*C. Text Encoder for sMRG*

The MRG module generates structured reports using the derived spinal assessments from the vision encoder: quantitative measurements, degenerative disease grading, and spinal disorders detection. These are injected into structured queries for clinical retrieval to enhance LLM outputs by retrieving relevant medical knowledge from vector database (VD) and knowledge graph (KG) databases and feeding it into context, thereby improving factual consistency and grounding. Unlike the traditional base RAG module, the proposed AgenticRAG integrates agentic reasoning loop instead of single-shot retrieval. The LLM judges relevancy and iteratively refines retrieval. The agent retrieves relevant information from two different knowledge sources: the first is a vector database, enabling semantic retrieval of document embeddings generated by the *mxbai-embed-large* model. The second is a temporal knowledge graph, which extracts triplets (node–relationship–node) from unstructured clinical text using predefined entities (e.g., MedicalCondition, AnatomicalStructure) and relationships (e.g., AFFECTS, LOCATED_IN) to find disease relationships among spinal pathologies, anatomical regions,

and diagnostic rules [18]. This dual retrieval strategy ensures comprehensive contextual retrieval by combining both semantic embeddings in addition to disease relational knowledge. In cases where retrieved information is irrelevant, the agentic reasoning continues iteratively to improve the relevancy of retrieved results. However, after multiple reasoning steps, if the retrieved information remains irrelevant, it defaults to generating the report based solely on measurements from vision encoders and the structured prompt template. Following the dual retrieval method, post-retrieval filtering refines the retrieved data by checking medical synonyms, aligning writing styles, and removing unhelpful cases based on doctor-defined grading rules. The refined retrieved information is then merged with level-based assessments to construct a comprehensive prompt that incorporates findings, impressions, recommendations, and structured level-based analysis. The final prompt is fed into the LLM generating detailed and clinically coherent sMRG. The novelty of the proposed hybrid AgenticRAG approach lies in its ability to combine real-time reasoning, adaptive retrieval, and structured prompt engineering, which generates reports that are not only comprehensive and precise but also aligned with clinical best practices and domain knowledge.

### D. Implementation Execution Settingss

The ECM model was implemented in PyTorch 2.7.0 and trained with the Adam optimizer, ReduceLROnPlateau scheduler (decay factor 0.9, patience 4), batch size of 16, initial learning rate of 1e-3, and trained for 150 epochs. Data augmentation included random sharpness, rotation, horizontal and vertical flips (with p=0.2). Hyperparameter configurations were derived from our previously established experimental setup [19]. Training was conducted on a workstation running Ubuntu 18.04.6 LTS with an Intel Core i9-7920X CPU, Titan V GPU (12GB VRAM), and 64GB RAM. For inference, the vision encoder and text encoder were executed on an NVIDIA GeForce RTX 5070 Ti. We employed Python 3.11.7, CUDA 11.8, and cuDNN 9, with LangChain for orchestration of the embedding, retrieval, and LLM-based generation. Additionally, ChromaDB was used for vector-based semantic retrieval, and Neo4j served as the backbone temporal knowledge graph database, together ensuring stable, efficient, and modular system performance.

### E. Evaluation Metrics

For image segmentation, we evaluated the performance of the vision encoder using the Dice Similarity Coefficient (DSC) [20]. For the sMRG evaluation, we use ROUGE [21], METEOR [22], FRUGAL, BLUE [23], and Bert score [24]. These evaluation metrics are applied to the findings section of the generated report and the physician's ground truth. Additionally, each AI-generated report is independently evaluated by three neurologists, who rate it on a scale from 1 (poor) to 10 (excellent).

## IV. RESULTS

### A. Segmentation

The proposed ECM is evaluated against state-of-the-art segmentation architectures on axial and sagittal MRI views. On sagittal T2-weighted scans (Table I), ECM achieved an average dice score of 97.58%, closely comparable with the best-performing model SegResNet (97.79%), while utilizing

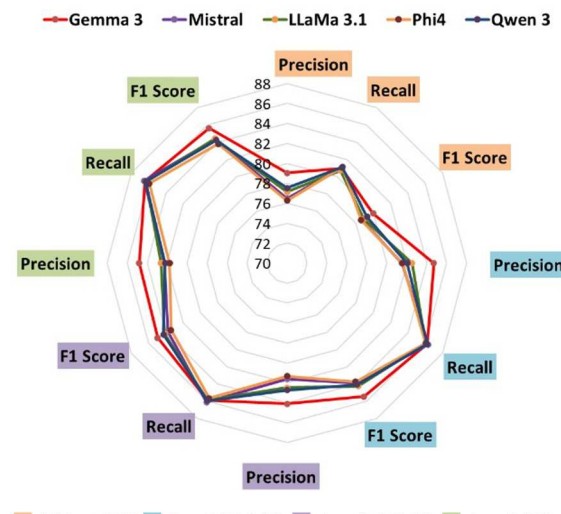

Fig.3 Ablation study for sMRG using the proposed AgenticRAG against the baselines of AgenticRAG_VD, AgenticRAG_KG, and without RAG.

only 9% of its architectural parameters. It achieved the highest performance of vertebrae segmentation with 96.64% Dice score and remains within 0.1-0.4 percentage of the top scores across other anatomical structures. On axial T1-weight scans (Table II), the model achieves top highest average dice score of 94.01% outperforming all other models. This consistent performance across different MRI views proof ECM's robustness and generalizability. Parameter efficiency (0.58M) compared to over 11.5M for UNETR, ECM provides an optimal balance between segmentation accuracy and computational efficiency. It also maintained real-time inference, processing at 24.28 frames per second with 3.62 GFLOPs, making it highly suitable for low-resource or time-constrained clinical environments. To assess 3D axial cross-projection accuracy, we assigned 100% to slices within the IVD and 0% outside, achieving 98.6% accuracy. Only one error (Patient#0259; L5-S1) occurred, with the slice near the IVD but over the sacrum cap, though neurologists confirmed no significant impact the AI report giving a score of 9/10.

### B. Medical Report Generation

To evaluate the proposed AgenticRAG for sMRG pipeline, we use five common LLMs: Gemma3 [25], Mistral [26], LLaMa3.1 [27], Phi4 [28], and Qwen3 [29]. Gemma3 achieved the top performance in ROUGE-L, Bert F1-Score and BLUE with 0.155, 0.856, and 0.176, respectively, while Mistral slightly outperforms in Meteor and FURGAL scores with 0.239, and 0.6175 as shown in Fig. 3. The proposed hybrid AgenticRAG demonstrates superior evaluation results for sMRG across all LLMs compared to the individual retrieval of AgenticRAG_KG and AgenticRAG_VD. A slight improvement observed in AgenticRAG_VD over AgenticRAG_KG, suggests that semantic information provides greater benefits than relational reasoning alone. Furthermore, Fig. 4 presents a structured generated report example for a single patient, highlighting the abnormal degeneration, spinal disorders, and misalignment conditions. While the second section provides a detailed level-based analysis for both vertebrae and IVDs, specifying normal and abnormal diagnosis for each level. The sMRG demonstrates the clinical value of detailed level-by-level analysis for the workflow efficiency. These highlighted sections indicate the

TABLE I. SEGMENTATION PERFORMANCE COMPARISON (DICE %) ON SAGITTAL MRI SCAN. ALL AI MODELS ARE TRAINED AND EVALUATED ON THE SAME DATASET, TRAINING ENVIRONMENT AND SETTINGS

| AI Segmentation Model | Sagittal MRI Scan (T2-weighted) | | | | | | | Params (Million) | FLOPs (G) | Frame/Sec |
|---|---|---|---|---|---|---|---|---|---|---|
| | PosterA | PosterB | Vertebrae | IVD | Sacrum | BG | Avg. | | | |
| DeepLabv3 | 99.07 | 99.02 | 96.43 | 94.84 | 94.29 | 99.67 | 97.35 | 58.7 | 125.5 | 43.19 |
| AttentionUnet | 98.83 | 98.78 | 96.41 | 94.77 | 93.79 | 99.55 | 97.33 | 5.5 | 728.3 | 13.08 |
| UNETR | 98.69 | 98.89 | 95.48 | 93.95 | 91.55 | 99.43 | 96.86 | 115.6 | 52.6 | 62.41 |
| SwinUNETR | 98.99 | 98.92 | 96.19 | 94.50 | 93.69 | 99.60 | 97.41 | 6.3 | 9.64 | 48.78 |
| SwinTransformer | 99.12 | 99.04 | 96.31 | 95.02 | 94.42 | 99.63 | 97.57 | 59.8 | 119.94 | 51.56 |
| SegResNet | **99.18** | **99.22** | 96.60 | **95.20** | **94.50** | 99.68 | **97.79** | 6.38 | 32.86 | 193.63 |
| TransUNet | 99.08 | 98.87 | 96.64 | 94.99 | 94.11 | **99.71** | 97.62 | 105.3 | 66.88 | 34.74 |
| **Proposed ECM** | 99.09 | 98.83 | **96.64** | 95.16 | 94.25 | 99.66 | 97.58 | **0.58** | **3.62** | **24.28** |

TABLE II. SEGMENTATION PERFORMANCE COMPARISON (DICE %) ON AXIAL T1 MRI SCAN. ALL AI MODELS ARE TRAINED AND EVALUATED ON THE SAME DATASET, TRAINING ENVIRONMENT AND SETTINGS

| AI Segmentation Model | Axial MRI Scan (T1-weighted) | | | | | | Params (Million) | FLOPs (G) | Frame/Sec |
|---|---|---|---|---|---|---|---|---|---|
| | IVD | PE | TS | AAP | BG | Avg | | | |
| DeepLabv3 | 97.81 | 93.09 | 94.26 | 78.02 | 99.70 | 92.69 | 58.7 | 125.5 | 43.19 |
| AttentionUnet | 96.53 | 92.24 | 93.71 | 77.99 | 99.64 | 92.02 | 5.5 | 728.3 | 13.08 |
| UNETR | 96.24 | 89.12 | 90.42 | 76.36 | 99.58 | 90.55 | 115.6 | 52.6 | 62.41 |
| SwinUNETR | 97.39 | 92.92 | 93.98 | 79.83 | 99.70 | 92.90 | 6.3 | 9.64 | 48.78 |
| SwinTransformer | 97.83 | 93.72 | 94.09 | 78.92 | 99.72 | 92.92 | 59.8 | 119.94 | 51.56 |
| SegResNet | **98.02** | **93.94** | 94.89 | 80.67 | 99.73 | 93.52 | 6.38 | 32.86 | 193.63 |
| TransUNet | 97.94 | 93.58 | 94.32 | 80.63 | 99.74 | 93.43 | 105.3 | 66.88 | 34.74 |
| **Proposed ECM** | 97.62 | 93.93 | **94.92** | **83.48** | **99.76** | **94.01** | **0.58** | **3.62** | **24.28** |

neurologists' validations of the AI-generated reports.
Clinical validation and expert oversight are essential to prevent over-reliance on automated reports. The neurologists assessed each report by correcting inaccurate statements, diagnosis gradings, or measurements. As shown in Fig. 4, each report was then rated on a scale from 1 (poor) to 10 (excellent). The overall average score across all 52 test cases

was 8 out of 10, indicating great factual alignment and minimal LLM hallucination.

## V. DISCUSSION

AutoSpineAI proposes to provide automatic lumbar spine analysis in terms of level-wise vertebrae and IVDs across the lumbar spine. It could balance the high efficiency and performance in medical MRI sagittal and axial image

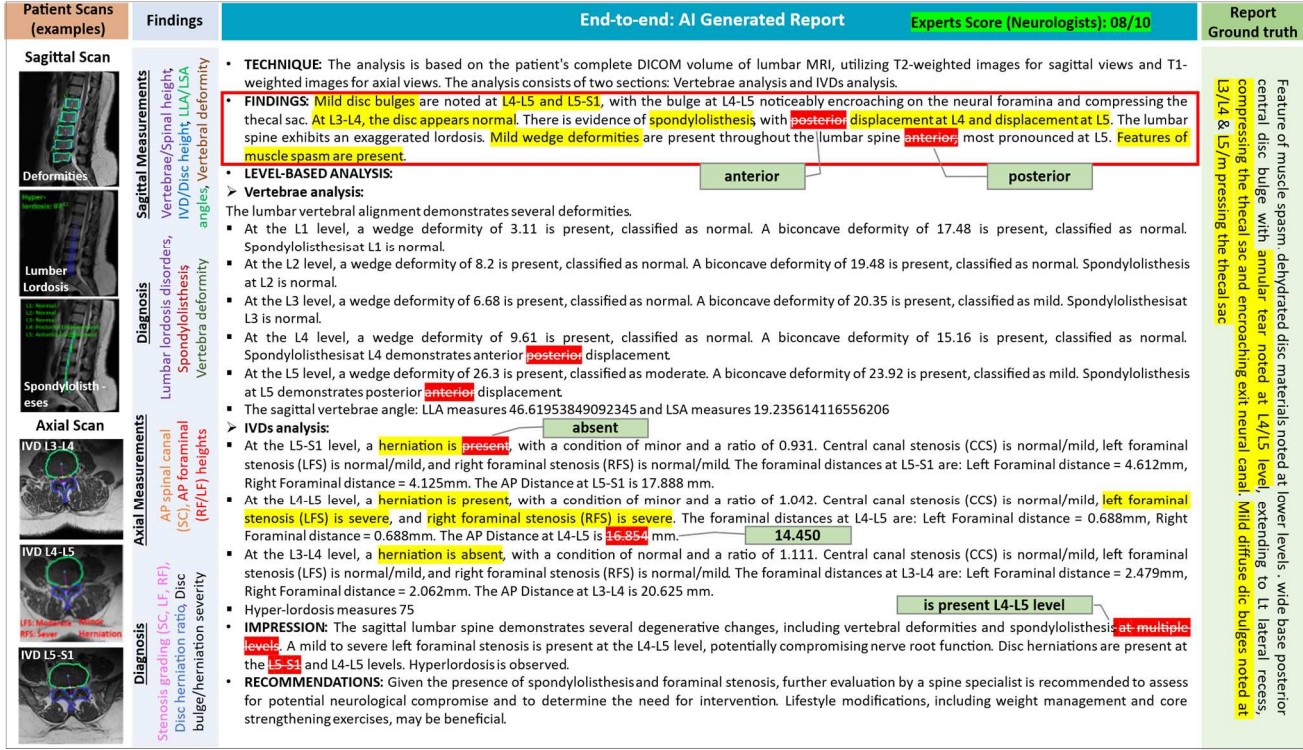

Fig.4. Structured medical report generated by the proposed MRG piepline, highlighting the overall findings section and level-based analysis for both vertebrae and IVDs. The neurologist's validations are highlighted by red and green pop-up corrected findings.

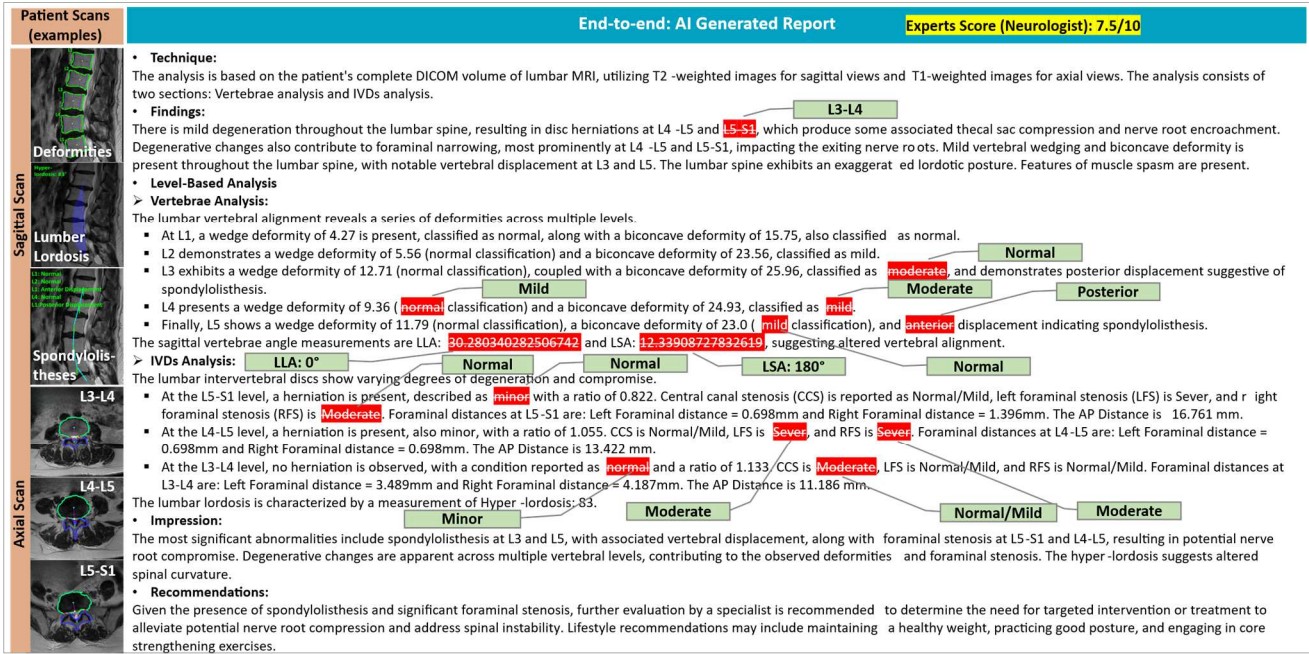

Fig.5. AI-generated lumbar spine report from our private real dataset as part of an ablation study, illustrating automated MRI vertebra and IVD analysis. The neurologists' validations are highlighted by red and green pop-up corrected findings.

TABLE III. ECM ATTENTION MECHANISMS IMPACT IN TERMS OF DICE AND INTERSECTION-OVER UNION (IoU) FOR AXIAL SCANS (SEE FIG. 2)

| Model Variant | CSA | PSA | SE | Bi-Mamba | Avg Dice (%) | IoU (%) |
|---|---|---|---|---|---|---|
| Baseline | ✗ | ✗ | ✗ | ✗ | 93.22 | 87.91 |
| +CSA | ✓ | ✗ | ✗ | ✗ | 93.77 | 88.79 |
| +PSA | ✗ | ✓ | ✗ | ✗ | 93.65 | 88.59 |
| +SE | ✗ | ✗ | ✓ | ✗ | 93.69 | 88.67 |
| +Bi-Mamba | ✗ | ✗ | ✗ | ✓ | 93.64 | 88.58 |
| +CSA+PSA | ✓ | ✓ | ✗ | ✗ | 93.82 | 88.87 |
| +CSA+PSA+SE | ✓ | ✓ | ✓ | ✗ | **93.97** | **89.08** |

TABLE IV. PERFORMANCE COMPARISON WITH OTHER MEDICAL REPORT GENERATION METHODS

| Method | ROUGE-L | METEOR | BLEU-1 | BLEU-2 | BLEU-3 | Data |
|---|---|---|---|---|---|---|
| AHP [13] | 0.285 | 0.154 | 0.400 | 0.250 | 0.172 | MIMIC-CXR |
| CMN [30] | 0.535 | 0.279 | 0.548 | 0.401 | 0.332 | Bladder pathology |
| R2Gen [31] | 0.277 | 0.142 | 0.353 | 0.218 | 0.145 | MIMIC-CXR |
| Gemma3 + AgenticRAG | 0.1733 | 0.166 | 0.25 | 0.017 | - | Lumbar Spine MRI |

TABLE V. COMPUTATIONAL COST ANALYSIS (SECOND) PER PATIENT OF EACH COMPONENT IN THE PROPOSED FRAMEWORK AS AN AVERAGE OVER ALL TESTING CASES

| Proposed RAG Approach | Vision Encoder | Document Retrieval | Agent Reasoning | Post Retrieval Cost time | Relational knowledge retrieval | Report on Generation | Total Cost |
|---|---|---|---|---|---|---|---|
| AgenticRAG_Hybrid | 5.61 | 2.07 | 2.82 | 3.40 | 0.24 | 16.48 | 30.62 |
| AgenticRAG_VD | | 2.00 | 2.57 | 3.05 | - | 16.38 | 29.61 |
| AgenticRAG_KG | | - | - | - | 0.24 | 15.75 | 21.6 |

segmentation and measurements via the vision encoder with only 0.58M parameters of ECM lightweight model where the current state-of-the-art models depend on large parameters (5.5M-115.6M). The ablation study shown in Table III demonstrates that each attention mechanism contributes positively to improving the segmentation performance: +CSA+PSA enhances by +0.60% Dice score and by +0.96% IoU. The final combination of CSA+PSA+SE achieves the best performance with 93.97% Dice score (+0.75%) and 89.08% IoU (+1.17%), representing improvement over the baseline. This validates our design choice of integrating many or multiple complementary attention mechanisms within the ECM architecture. For the text Encoder side, the proposed hybrid AgenticRAG approach with the Gemma3 LLM provides the best performance for sMRG as shown in Fig. 3. The dual-retrieval framework reduces hallucinations by grounding generations into factual, retrieved information. Table IV compares the performance of automated medical report generation systems against other clinical modalities. The performance drop in lumbar spine MRI report generation is mainly due to the lack of large public datasets and the complexity of interpreting both axial and sagittal views, highlighting the need for domain-specific modeling. Moreover, as the dataset primarily consists of patients with symptomatic low back pain from a single clinical source, it may limit model generalizability across diverse populations, highlighting the need for expanded data resources. AutoSpineAI is intended to support clinicians rather, as relying solely on automated reports without expert oversight poses significant risks in critical medical diagnoses. Future work will involve thorough usability testing with radiologists to evaluate its integration into clinical workflows. For cost commutation, Table V summarizes the computational cost across each component of the proposed CAD system. The full

report generation method takes an average of 30.62 seconds per patient, with only 5.61 seconds for segmentation and spinal measurements. As an ablation study using a separate private MRI real dataset with approved IRB, AutoSpineAI demonstrates its ability to perform comprehensive MRI-based LSS analysis and generate descriptive reports by leveraging both image and text encoders. In the absence of ground truth textual reports, neurologists evaluated the AI-generated outputs and assigned an impressive average score of 7.5 out of 10 across 100 patient cases. Fig. 5 presents a complete example of sagittal and axial MRI analysis, showcasing the automatic generation of detailed AI reports without any user intervention.

## VI. Conclusion

AutoSpineAI introduces a lightweight, multimodal framework for assisting in lumbar spine MRI analysis and automated report generation. Its Efficient Contextual Module (ECM) achieves high segmentation accuracy while maintaining low computational demand, making it suitable for use in settings with limited resources. For report generation, the hybrid AgenticRAG approach combines semantic understanding with structured domain reasoning to deliver precise, vertebral-level diagnostic outputs. Extensive evaluations demonstrate the system's robustness, adaptability, and clinical relevance. AutoSpineAI effectively integrates visual and textual features to generate structured reports within 30 seconds, highlighting its potential as a reliable and efficient tool for enhancing diagnostic workflows in spinal radiology.

## Acknowledgment

This work was supported by the National Research Foundation of Korea (NRF) grant funded by the Korean government (MSIT) (No. RS-2023-00256517) and by the TUBITAK (The Scientific and Technological Research Council of Turkey) under Grant Number: 123N325. This work was supported by the IITP(Institute of Information & Communications Technology Planning & Evaluation)-ITRC(Information Technology Research Center) grant funded by the Korea government (Ministry of Science and ICT) (IITP-2025-RS-2024-00437191).

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
