# OpenReview forum: "AutoSpineAI: Lightweight Multimodal CAD Framework for Lumbar Spine MRI Assessments"
_IEEE.org/EMBS/BHI/2025/Conference — BHI 2025_

### Official Review · Reviewer_QErY · 2025-06-30
**AutoSpineAI: Lightweight Multimodal CAD Framework for Lumbar Spine MRI Assessments**

**Confidence:** 5
**Clarity Of Writing:** good
**Clinical Significance:** great
**Methodological Novelty:** great
**Overall Rating:** 7

**Experiments And Results:**

good

**Questions For The Authors:**

1. Clearly define Retrieval-Augmented Generation (RAG) in the manuscript to ensure clarity for readers unfamiliar with the concept.

2. The font sizes in Figure 3 are too small to be legible. Additionally, consider using a more effective highlighting method to enhance visual clarity and emphasis.

3. Provide a discussion of potential biases present in the dataset, including how they may affect model performance and generalizability.

4. Strengthen the failure analysis by offering a more comprehensive and detailed error analysis for both the segmentation and report generation components.

5. Address the clinical risks, limitations, and possible sources of error associated with the proposed approach, particularly in real-world deployment scenarios.

6. While the paper introduces AgenticRAG, it does not adequately explain how it differs from standard RAG in terms of architectural design or logical workflow. A clearer distinction is necessary.

7. The pipeline’s reliance on the mid-sagittal slice for segmentation and axial localization warrants further clarification. Specifically, discuss the robustness of this approach across anatomical variability and whether off-midline pathologies may be overlooked.

8. The methodology for extracting spinal measurements (e.g., angles, herniation ratios) is only briefly mentioned. Please elaborate on the computation techniques and provide information on measurement variability relative to manual expert assessments.

9. Although the reported runtime statistics are promising, additional context is needed. For instance, clarify whether the 30-second runtime refers to end-to-end processing per patient or per image, and whether it includes pre-processing steps.

**Strengths:**

1. Combines sagittal and axial MRI data, quantitative spinal measurements, and clinical text to provide a comprehensive, multimodal diagnosis

2. Use of agentic retrieval-augmented generation (RAG) method, combining semantic vector search and knowledge graph-based reasoning, leads to high-quality, structured, and clinically relevant diagnostic reports.

3. With a full inference time of 30 seconds and low computational demands, this method is deployable in low-resource environments.

**Summary Of The Paper:**

Authors have proposed a lightweight, multimodal AI framework designed for automated analysis of lumbar spine MRIs and structured medical report generation. It features an efficient segmentation model (ECM) that achieves high accuracy with minimal computational cost. The system extracts detailed spinal measurements and uses an LLM-based AgenticRAG method to generate clinically coherent, level-specific reports in under 30 seconds. AutoSpineAI offers a scalable, accurate, and practical solution for improving diagnostic workflows in spine radiology.

**Weaknesses:**

1. Model is trained and validated on a relatively small, public lumbar spine MRI dataset.

2. Has not been tested on independent or external datasets.

3. The system needs to be evaluated deeply for failure modes or where the system may underperforms.

4. AgenticRAG method (new) needs broader validation across different domains.

---

### Official Review · Reviewer_fyyA · 2025-07-14
**AutoSpineAI, an innovative computer-aided diagnosis (CAD) framework that performs automated segmentation, structured spinal measurement extraction, and structured medical report generation (sMRG) for lumbar spine MRI**

**Confidence:** 3
**Clarity Of Writing:** great
**Clinical Significance:** great
**Methodological Novelty:** excellent
**Overall Rating:** 7

**Experiments And Results:**

great

**Questions For The Authors:**

What is the size and diversity of your MRI dataset? Is it publicly available?
Has the system been tested on data from other scanners, centers, or spinal pathologies?
How do you ensure factual consistency in the generated medical reports? Was hallucination evaluated?
What clinical ontologies or terminologies are used in your knowledge graph for structured retrieval? Were they verified by experts?
Have you conducted usability testing with radiologists or physicians? What is the average time saved in workflow?

**Strengths:**

Combines image segmentation with LLM-based text generation in a seamless diagnostic pipeline. The structured prompting of a domain-informed LLM with retrieval capabilities enables more accurate and tailored report generation. The pipeline mirrors radiologist workflow: segmentation → measurement → interpretation → report.

**Summary Of The Paper:**

The paper addresses the clinical burden of manual MRI interpretation for CLBP and lumbar spine disorders. It automates the pipeline from 3D MRI image preprocessing to medical report generation using AI. Authors extract mid-sagittal and localized axial slices from 3D DICOM MRIs using a 3D cross-projection mechanism and use a custom lightweight CNN (ECM) with attention modules to segment vertebrae and intervertebral discs. They derive quantitative structural spinal measurements (SSM) including disc degeneration grades and anomalies.
Authors pass structured measurements as prompts to a Gemma3-based LLM to generate structured diagnostic reports using a knowledge-aware agentic RAG approach.
They achieved high segmentation accuracy: Dice score of 97.58% (sagittal) and 94.01% (axial). Report generation scored 83.51% BERT F1, 19.33% METEOR, and 15.31% ROUGE-1, suggesting clinically coherent language outputs.

**Weaknesses:**

The paper lacks detailed statistics on the dataset used (e.g., number of scans, patient diversity, tumor subtypes).
No indication of testing on independent or multi-institutional datasets. LLMs may hallucinate or produce inconsistent outputs—this is not addressed. It is unclear how AutoSpineAI's reports compare to radiologist-generated ones in clinical accuracy or readability.

---

### Official Review · Reviewer_iVEM · 2025-07-16
**Promising AgenticRAG approach with efficient segmentation design**

**Confidence:** 3
**Clarity Of Writing:** good
**Clinical Significance:** good
**Methodological Novelty:** great
**Overall Rating:** 7
**Final Rating:** 8

**Experiments And Results:**

good

**Questions For The Authors:**

1. How much does the dual retrieval approach improve performance over simpler baselines? Understanding the quantitative benefit of AgenticRAG versus basic RAG or vector-only retrieval would help assess whether the added complexity provides meaningful value. This could significantly impact the evaluation of your methodological contribution.
2. What is the accuracy of your 3D axial cross-projection localization? This step seems critical to the pipeline success. How robust is it across different spine curvatures, patient anatomies, and imaging protocols? Understanding failure rates and handling of challenging cases would provide important insights into system reliability.
3. Which components of your ECM architecture contribute most to the efficiency gains? Ablation studies showing the relative importance of the different attention mechanisms (CSA, PSA, SE, Bi-Mamba) would help understand which innovations are essential versus architectural choices.
4. How do you handle cases where the AgenticRAG system can't find relevant information? Understanding the system's behavior in edge cases and the confidence measures provided to clinicians would be valuable for assessing clinical safety and utility.

**Strengths:**

1. Impressive parameter efficiency with maintained performance: The ECM model achieves competitive segmentation performance (97.58% Dice score) with only 0.58M parameters, significantly fewer than state-of-the-art models (5.5M-115.6M), making it highly suitable for resource-constrained clinical environments. This represents a genuine advancement in efficient medical AI architecture design.
2. Novel multimodal integration approach. The combination of sagittal and axial MRI views with clinical text for comprehensive spinal diagnosis is well-motivated and technically sound. The 3D axial cross-projection method for automatically localizing corresponding slices is innovative and addresses a real clinical workflow challenge.
3. Pioneering work in spine report generation: The AgenticRAG approach combining semantic retrieval with knowledge graph-based reasoning represents a significant methodological contribution to medical report generation. The dual-retrieval strategy shows clear architectural advantages, and the 83.51% BERT F1-Score demonstrates strong semantic similarity with ground truth reports.
4. Comprehensive end-to-end clinical solution: The paper addresses the complete pipeline from MRI image processing to structured report generation, which is clinically relevant and practical. The 30-second inference time makes it deployable in real clinical workflows.
5. Strong technical evaluation: The comparison against multiple established segmentation baselines (DeepLabv3, UNETR, SwinUNETR, etc.) provides solid evidence for the architectural contributions, and the NLG metrics are competitive with other medical domains despite the inherent complexity of spine reporting.

**Summary Of The Paper:**

The paper presents AutoSpineAI, a fully automated computer-aided diagnosis (CAD) framework for lumbar spine MRI analysis and structured medical report generation. The system processes 3D MRI DICOM volumes by extracting mid-sagittal slices for vertebrae and intervertebral disc (IVD) segmentation, then uses a 3D axial cross-projection method to localize corresponding axial slices. The framework introduces an Efficient Compact Model (ECM) for lightweight segmentation and employs an agentic LLM-driven retrieval approach combining semantic information with knowledge graph-based reasoning to generate detailed diagnostic reports. The system achieves Dice scores of 97.58% and 94.01% for sagittal and axial segmentation respectively, and generates structured reports within 30 seconds with 83.51% BERT F1-Score.

**Weaknesses:**

1. Knowledge graph maintenance strategy needs development: While you've built an impressive dual-retrieval system, medical knowledge evolves rapidly - treatment protocols change, new diagnostic criteria emerge, imaging guidelines get updated. Without a framework for keeping the knowledge graph current, this becomes a static system that will degrade over time. Even outlining basic principles for periodic updates would strengthen the long-term practical value.
2. Limited generalizability assessment: The single-institution dataset of 515 patients, while substantial for spine imaging, cannot capture the variability of different MRI protocols, patient populations, and clinical settings. The robustness across different imaging parameters and demographic groups remains unclear, which could impact real-world deployment.
3. Clinical workflow integration unaddressed: While the technical performance is strong, the paper doesn't address how this system would integrate into existing radiology workflows, handle edge cases, or provide confidence measures for clinical decision-making. Understanding failure modes and system limitations would be valuable for practical deployment.

---

### Official Review · Reviewer_Zt2w · 2025-07-18
**Thoughtful choice of health application, but novelty of proposed tool is unclear**

**Confidence:** 3
**Clarity Of Writing:** poor
**Clinical Significance:** good
**Methodological Novelty:** poor
**Overall Rating:** 2

**Experiments And Results:**

fair

**Questions For The Authors:**

A central part of the contribution described in this paper is the ECM model for image segmentation. How does this model connect to existing work? (For example, when I look up ECM I find a similarly-titled preprint, “ECMNet:Lightweight Semantic Segmentation with Efficient CNN-Mamba Network” which seems may describe a similar network architecture concept). What is novel about the proposed ECM net in this work compared to prior literature and preprints, and what is the rationale for any novel design decisions? Have the authors considered publishing on this parameter-efficient net separately?

Have the authors considered segmentation or interpretation of the three-dimensional scan directly, without taking slices?

**Strengths:**

The study identifies a lesser-addressed medical problem in the image segmentation field, that of lumbar spine MRI for chronic lower back pain. To that end, this study worked with a dataset of 515 lumbar spine MRI scans. It is important for the field of health AI to address a variety of real-world medical conditions, and this paper provides an important contribution to the diversity of applications of health AI.

Model performance was very good, especially on the segmentation task, with a dice score above 97%. Model inference time is reported (30 seconds) and seems reasonably timely.

**Summary Of The Paper:**

This study proposes a fully automated pipeline for lumber spine MRI analysis, with full MRI as input and structured medical report as output. The pipeline first extracts slices of the MRI scan for automated segmentation. Next, a parameter-efficient network architecture performs segmentation. Finally, a large language model is used to write a medical report of the results.

**Weaknesses:**

The writing style of this paper is difficult to follow. The abstract does not clearly articulate the key points of the paper, and the full text would also benefit from language editing for clarity and style. Introduction of acronyms and key terms is inconsistent. Figures, such as Figure 1, are crowded and difficult to parse.

Introduction of a tool name, “AutoSpineAI”, is too ambitious relative to what is delivered. The AutoSpineAI pipeline first consists of a segmentation model – this ECM segmentation model may deserve a name if sufficiently novel, but the novelty seems unclear (see questions for authors). The second half of the pipeline, use of an LLM to generate written output reports, tests the application of several off-the-shelf models, which is more consistent with benchmarking than new tool development. There is also no mention of deploying the proposed tool as a software or public repository to facilitate community development and use.

---

### Official Review · Reviewer_zCwC · 2025-07-20
**AutoSpineAI: Lightweight Multimodal CAD Framework for Lumbar Spine MRI Assessments.**

**Confidence:** 3
**Clarity Of Writing:** good
**Clinical Significance:** great
**Methodological Novelty:** great
**Overall Rating:** 7

**Experiments And Results:**

great

**Questions For The Authors:**

Can you provide an ablation study or analysis on how each attention mechanism or AgenticRAG component contributes to the overall performance of the model?
Have you evaluated or considered the robustness of the LLM-generated reports? For instance, what happens if the input measurements are slightly perturbed?

**Strengths:**

1. Novel and Efficient Architecture:
The proposed ECM segmentation model combines channel, patch, and bidirectional attention mechanisms in a compact encoder-decoder framework, achieving high segmentation accuracy (Dice: 97.58% sagittal, 94.01% axial) while maintaining real-time performance.
2. AgenticRAG for Medical Report Generation:
The integration of vector-based semantic retrieval with graph-based disease knowledge in the report generation module is a novel and impactful approach. It ensures more contextual and clinically valid reports by combining factual reasoning with domain knowledge.
3. Multimodal, End-to-End Design:
The AutoSpineAI system is end-to-end, supporting a fully automated pipeline from MRI input to structured report output, which addresses a real clinical bottleneck in radiology workflows.
4. Clinical Relevance:
By capturing both sagittal and axial information and reporting at a vertebra-level granularity, the tool aligns well with radiology practices and has strong potential for clinical translation.
5. Computational Efficiency:
The proposed method achieves performance comparable to or better than large models with a fraction of the parameters and FLOPs, which is critical for resource-constrained environments.

**Summary Of The Paper:**

The paper proposes AutoSpineAI, a lightweight, multimodal computer-aided diagnosis (CAD) system for analyzing lumbar spine MRI scans and generating structured medical reports (sMRG). The framework consists of:
•	A lightweight segmentation model (Efficient Compact Model, ECM) for sagittal and axial MRI views.
•	A 3D axial projection mechanism to align and extract relevant slices across views.
•	A Spinal Structure Measurements (SSM) module to quantify structural abnormalities.
•	An Agentic Retrieval-Augmented Generation (AgenticRAG) system using semantic and graph-based knowledge retrieval to generate clinically coherent, level-wise diagnostic reports via LLMs.

The authors utilize a publicly available lumbar spine dataset comprising sagittal (T2) and axial (T1) 3D MRI volumes, annotated by experts.

Experimental evaluation demonstrates that the ECM achieves state-of-the-art segmentation accuracy with minimal computational cost (only 0.58M parameters). Report generation achieves competitive metrics, including an 83.51% BERT F1-score within an average of 30 seconds, showing promise for deployment in time-sensitive clinical settings.

**Weaknesses:**

1. Language and Presentation Issues:
The manuscript contains multiple grammatical errors, awkward phrasings, and formatting inconsistencies (e.g., "sever" vs. "severe", inconsistent figure/table placements). These detract from clarity and professionalism.
2. Lack of Ablation Study:
There is no ablation analysis of the ECM model components (e.g., attention layers, Bi-Mamba) or AgenticRAG components (vector-only, graph-only, and hybrid). This limits insight into the contribution of each element.
3. Insufficient Discussion of Limitations:
Although the discussion touches on dataset scarcity, it does not mention potential LLM hallucinations, the need for clinical validation, or the risks of over-reliance on automated reports.